# Towards harmonization of image velocimetry techniques for river surface velocity observations

Matthew. T. Perks[1], Silvano Fortunato Dal Sasso[2], Alexandre Hauet[3], Elizabeth Jamieson[4], Jérôme Le Coz[5], Sophie Pearce[6], Salvador Peña-Haro[7], Alonso Pizarro[2], Dariia Strelnikova[8], Flavia Tauro[9], James Bomhof[4], Salvatore Grimaldi[9,10], Alain Goulet[4], Borbála Hortobágyi[1], Magali Jodeau[11,12], Sabine Käfer[13], Robert Ljubičić[14], Ian Maddock[6], Peter Mayr[15], Gernot Paulus[8], Lionel Pénard[5], Leigh Sinclair[4], and Salvatore Manfreda[16]

[1]School of Geography, Politics and Sociology, Newcastle University, Newcastle upon Tyne, United Kingdom.
[2]Department of European and Mediterranean Cultures: Architecture, Environment and Cultural Heritage (DiCEM), University of Basilicata, 75100 Matera, Italy.
[3]Electricité de France, DTG, Grenoble, France.
[4]National Hydrological Services, Environment and Climate Change Canada.
[5]INRAE, UR RiverLy, River Hydraulics, Villeurbanne, France.
[6]School of Science and the Environment, University of Worcester, Worcester, UK.
[7]Photrack AG: Flow Measurements, Ankerstrasse 16a, 8004 Zürich, Switzerland.
[8]School of Geoinformation, Carinthia University of Applied Sciences, 9524 Villach, Austria.
[9]Department for Innovation in Biological, Agro-food and Forest Systems, University of Tuscia, Viterbo, Italy.
[10]Department of Mechanical and Aerospace Engineering, Tandon School of Engineering, New York University, Brooklyn, NY, United States.
[11]Electricité de France, R&D, Chatou, France
[12]LHSV, Chatou, France.
[13]Verbund Hydro Power GmbH, 9500 Villach, Austria.
[14]Faculty of Civil Engineering, University of Belgrade, Belgrade 11120, Serbia.
[15]flussbau iC, 9500 Villach, Austria.
[16]Department of Civil, Architectural and Environmental Engineering, University of Naples Federico II, Via Claudio 21, 80125 Napoli, Italy.

**Correspondence:** Matthew T. Perks (matthew.perks@newcastle.ac.uk)

**Abstract.** Since the turn of the 21[st] Century, image based velocimetry techniques have become an increasingly popular approach for determining open-channel flow in a range of hydrological settings across Europe, and beyond. Simultaneously, a range of large-scale image velocimetry algorithms have been developed, equipped with differing image pre-processing, and analytical capabilities. Yet in operational hydrometry, these techniques are utilised by few competent authorities. Therefore, imagery collected for image velocimetry analysis, along with reference data is required both to enable inter-comparisons between these differing approaches and to test their overall efficacy. Through benchmarking exercises, it will be possible to assess which approaches are best suited for a range of fluvial settings, and to focus future software developments. Here we collate, and describe datasets acquired from seven countries across Europe and North America, consisting of videos that have been subjected to a range of pre-processing, and image velocimetry analysis (Perks et al., 2020, http://doi.org/10.4121/uuid: 014d56f7-06dd-49ad-a48c-2282ab10428e). Reference data is available for 12 of the 13 case studies presented enabling these data to be used for reference and accuracy assessment.

# 1 Introduction

When designing hydrological monitoring networks, or acquiring opportunistic measurements for determining open-channel flow, the optimum choice of apparatus is likely to be a compromise between the data requirements, resource availability, and the hydro-geomorphic characteristics of the site (Mishra and Coulibaly, 2009). Generally, hydro-geomorphic factors will

include: channel width and depth, the range of flow velocities, presence of secondary circulation, and cross-section stability. Each field measurement technique will have a designed range of optimum operating conditions, under which, robust flow measurements should be expected (e.g. ISO 24578:2012). However, under conditions beyond their designed operating range, greater levels of uncertainty will ensue. This may therefore preclude certain approaches for deployment under very shallow, or flood flow conditions for example. Logistical and practical constraints may also limit the deployment of apparatus. For

example, techniques that require the device to be in contact with the water during operation may not be feasible for health and safety reasons during periods of high-flow, or due to staff availability (Harpold et al., 2006). As a result of some of these challenges, the potential for implementing alternative, non-contact approaches has been recently explored. Within this field of research, image velocimetry has emerged as an exciting new approach for determining a key hydrological characteristic, namely flow velocity.

Image velocimetry involves the application of cross-correlation, or computer vision techniques on a series of consecutive images (or extracted video frames) to generate vectors of water velocities across a field-of-view. It was originally developed for use in highly controlled laboratory settings. However, since its original conception, its application has expanded from use in the laboratory (e.g., Dudderar and Simpkins, 1977; Adrian, 1984; Pickering and Halliwell, 1984), to include a wide variety of experimental conditions. Most notably it has been deployed outside of the controlled environment of the laboratory and

into the domain of the field scientist (e.g., Fujita et al., 1998). It is now applied in complex environments including situations where lighting is not controlled, the camera platform may be mobile (e.g. on unmanned aerial systems (UAS)), images may be acquired oblique to the direction of flow, and at an angle that changes over time (e.g. Detert and Weitbrecht, 2015; Tauro et al., 2016b; Perks et al., 2016).

This technique is also becoming increasingly popular with the wider hydrological community (Tauro et al., 2018a), and this

has been aided by two key factors. The first of which is the development of platforms and hardware that enable high-definition images and videos to be captured precisely, stored, and transferred to locations where image processing can occur. Secondly, many researchers utilizing image velocimetry techniques have chosen to develop their own specific processing capabilities, leading to the development of a range of both open-source and proprietary software for image pre-processing, and velocimetry analysis (Table A1). Whilst this has led to a breadth of options for researchers conducting image velocimetry analysis, inter-

comparisons of their efficacy under a range of environmental settings, and flow regimes is currently lacking (Pearce et al., 2020). Therefore, there is an urgent need to comprehensively understand and appreciate limitations of the differing image velocimetry approaches that are available to the scientific community.

Here, we present a range of datasets that have been compiled from across seven countries in order to facilitate these inter-comparison studies (Figure 1, Perks et al. (2020)). These data have been independently produced for the primarily purposes

of: (i) enhancing our understanding of open-channel flows in diverse flow regimes; and (ii) testing specific image velocimetry techniques. These datasets have been acquired across a range of hydro-geomorphic settings, using a diverse range of cameras, encoding software and controller units. Image sequences have then been subjected to a range of differing image pre-processing steps using a range of image processing software. The compilation of these diverse datasets offers the research community a resource for addressing key challenges that have been identified in the use of image velocimetry algorithms. These include but are not limited to the potential for the characteristics of the seeding material (e.g. particle density) to affect the resultant velocity estimates (Dal Sasso et al., 2018; Pizarro et al., 2020); the impact of UAS movement on velocity measurements (Lewis and Rhoads, 2018), and testing of different image stabilisation approaches to address this. Additional assessments may concern the role of image pre-processing (e.g. background suppression; Thielicke and Stamhuis, 2014), and the role of pixel resolution and video length on errors under differing flow conditions (Tauro et al., 2018b).

## 2 Experimental Design

Given the range of image velocimetry techniques that have been developed in recent years, benchmarking datasets covering a range of hydro-geomorphic conditions and acquisition platforms are required in order to test the accuracy and precision of each algorithm for the determination of 1- and 2-dimensional surface velocities. The examples that we describe in this section have been acquired by a range of platforms including UAS, fixed and hand-held cameras. The geographical characteristics of the sites are also widely varied. Catchment areas span $20$–$17\,460\,\mathrm{km^2}$, captured channel widths range from $5$–$59\,\mathrm{m}$, minimum flow depth is $0.10\,\mathrm{m}$, with a maximum of over $7\,\mathrm{m}$, and mean flow velocities range from $0.13$ to over $6\,\mathrm{m\,s^{-1}}$. Where possible, reference data generated by established and widely accepted approaches (e.g. current meter, and acoustic Doppler current profiler (ADCP)) have been collected simultaneously, or at the same river stage as images are acquired. The details pertaining to the hydrological conditions of deployment, configuration of the camera setup, pre-processing of imagery, and where relevant, details of published results from these datasets are presented in the following sections and summarised in Table A2.

### 2.1 River Arrow, UK

On 1st November 2017, a field experiment was undertaken on the River Arrow in Warwickshire (UK), to ascertain the accuracy of two differing image velocimetry approaches. The location of this experiment was in the mid-reaches of the catchment with a contributing area of $94\,\mathrm{km^2}$. This is a stable, meandering section of the river with an approximate width of $5\,\mathrm{m}$. During the experiment, mean depth and velocity were $0.22\,\mathrm{m}$ and $0.42\,\mathrm{m\,s^{-1}}$ respectively and water turbidity was minimal with the gravel bed being clearly visible in the footage.

The two deployments differed as both fixed (bankside pole-mounted) and mobile (UAS) imaging systems were used. Footage acquired from these two systems was captured concurrently, permitting direct comparisons to be made between the two. The mobile imaging system consisted of a DJI Phantom 4 Pro UAS equipped with a 1" camera CMOS sensor. This was used to collect nadiral footage with the camera's y-axis orthogonal to the direction of flow. Video was collected by the UAS for $4\,\mathrm{min}\,18\,\mathrm{s}$ whilst hovering at an elevation of approximately $20\,\mathrm{m}$ over the field of interest (Figure 2a). Footage was recorded

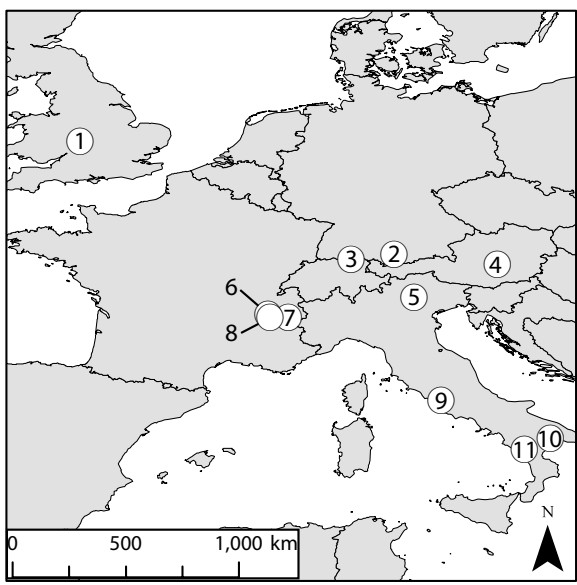

**Figure 1.** Locations of the monitoring sites from which data is presented. Numbers correspond with the in-text subsections: (1) River Arrow, UK; (2) River Thalhofen, Germany; (3) Murg River, Switzerland; (4) Alpine River, Austria; (5) River Brenta, Italy; (6) La Morge, France; (7) St-Julien torrent, France; (8) River La Vence, France; (9) River Tiber, Italy; (10) River Bradano, Italy; (11) River Noce, Italy. Not shown: (12) Castor River, Canada; (13) Salmon River, Canada. Map spatial reference: ETRS (1989).

at a pixel resolution of $1920 \times 1080$, and frame rate of $30\,\mathrm{Hz}$. The second approach consisted of a GoPro Hero 4 mounted at an oblique angle on a stationary telescopic pole at a height of approximately $2\,\mathrm{m}$ above the water surface. Video footage was simultaneously collected for $5\,\mathrm{min}\,37\,\mathrm{s}$ at a pixel resolution of $1920 \times 1080$, and frame rate of $30\,\mathrm{Hz}$. During the period of recording, sequences consisting of both unseeded flow, and artificially seeded flow are visible. For the seeded element, cornstarch ecofoam chips were added to the water surface immediately upstream of the area of interest. These tracers are clearly visible in the footage and are distributed evenly in the cross-section. Seeding was carried out to enhance the availability of traceable features in the low-flow conditions.

From the recordings, datasets each consisting of 99 consecutive images (sampled at a frame rate of $5\,\mathrm{Hz}$, and converted to grayscale intensity) were extracted from both the UAS and GoPro footage under both seeded and unseeded conditions. As a result of camera movement for both the UAS and GoPro footage, image sequence stabilisation was carried out using Fudaa-LSPIV (Table A1). In order to enable the conversion of pixel units to metric units a total of ten ground control points (GCPs), which were visible throughout the duration of the video, were distributed across both banks (Figure 2a). These GCPs were surveyed and their positions utilised for image orthorectification using Fudaa-LSPIV (Table A1). Subsequently, the orthorectified images have a scaling of $0.0174\,\mathrm{m}\,\mathrm{px}^{-1}$ (Figure 2b).

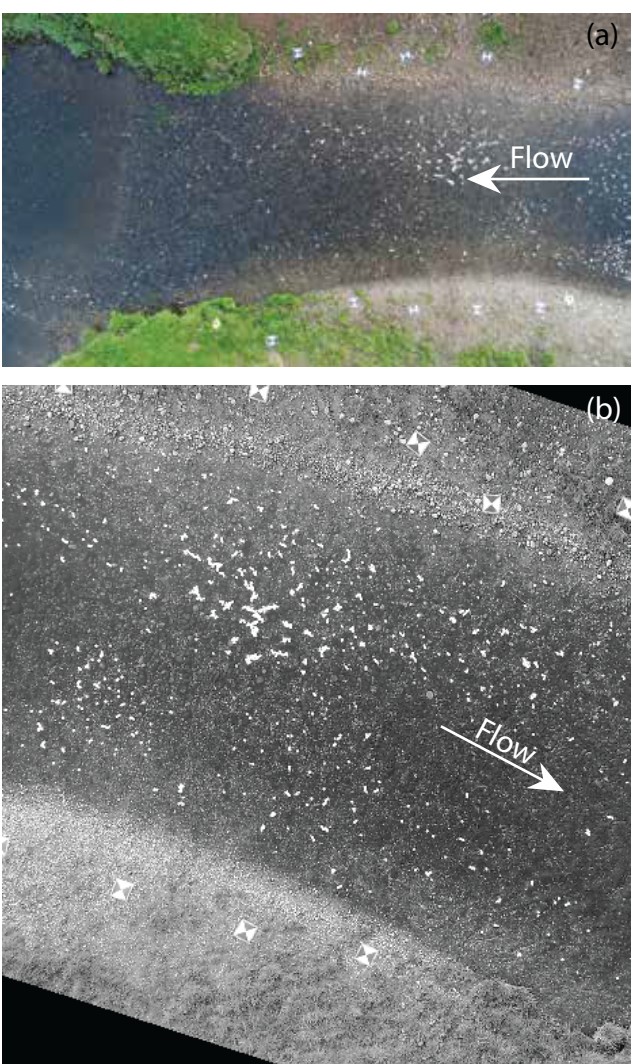

**Figure 2.** (a) Footage acquired by the Phantom 4 UAS over the River Arrow, and (b) following orthorectification and grayscale conversion. The Ecofoam chips and ground control points are clearly visible in both images. The direction of flow is indicated by the arrows.

Reference data was obtained through the deployment of a Valeport 801 electromagnetic current meter. Measurements were made for a period of $30\,\text{s}$ just below the water surface with the time-averaged value being reported. Measurements were obtained for five cross-sections spaced approximately $1.5-2\,\text{m}$ apart, within which, 9–10 individual measurements were obtained with a spacing of $0.5\,\text{m}$ between each. The location of each measurement is provided in pixel units based on the stabilised and orthorectified imagery of the UAS and GoPro.

## 2.2 River Thalhofen, Germany

On $27^{th}$ July 2017, a Vivotek IB836BA-HT network surveillance camera was utilised to capture footage for image velocimetry analysis on the River Thalhofen in Germany. At the time of deployment, the river width was approximately $28\,m$, the river stage was $1.45\,m$, and ADCP derived discharge and mean velocity were $52.515\,m^3\,s^{-1}$ and $1.7\,m\,s^{-1}$ respectively. The camera was fixed in location with the camera lens at an approximate angle of $25°$ from nadir and the image y-axis approximately $5°$ from being perpendicular to the direction of flow. Images were collected for a duration of $2\,s$ at a resolution of $1280 \times 800$px and frame rate of $30\,Hz$. Despite the presence of highly turbid water, which can diminish contrast across the water surface, the presence of highly visible turbulent structures advecting downstream offers the potential for the extraction of surface velocity information from these images. Image pre-processing consisted of orthorectification (using Photrack software (Table 1A)), and color conversion to gray-scale. 56 consecutive images which have been subjected to these processing steps are presented here at their original frame rate of $30\,Hz$. The pixel dimensions of the processed imagery are $0.01\,m$ in the x and y-axis.

Reference data was acquired by means of a Teledyne RiverPro ADCP and consists of a single transect consisting of 280 measurements along the cross-section with an average spacing of $0.09\,m$. ADCP data was acquired with a bin-depth of $0.06\,m$ with the upper-most measurement occurring at a distance of $0.22\,m$ below the water surface.

## 2.3 Murg River, Switzerland

On the $6^{th}$ April 2016, aerial surveys were undertaken in order to acquire imagery for determining the bathymetry, surface velocity, and to subsequently derive the flow discharge of the Murg River, Switzerland (Detert et al., 2017). The experiment took place in the middle reaches of the catchment, with a contributing area of $212\,km^2$. The experimental reach was a stable, straight section totalling $75\,m$ in length, along which, the water depth was approximately $0.35\,m$ and channel width was $12\,m$. The discharge at the time of survey was $2.76\,m^3\,s^{-1}$. For the aerial survey a DJI Phantom FC40 was deployed at a stable altitude of $30\,m$ to track the movement of artificial tracers throughout the reach. The UAS was equipped with a GoPro Hero3+ black edition 4K camera, capable of capturing a large spatial footprint whilst deployed at a relatively low altitude. However, this also generates a considerable barrel distortion effect which must be overcome during image processing. This system was used to collect nadiral footage with the camera's y-axis perpendicular to the flow direction. Video footage was acquired for a period of $2\,min\,11\,s$ at a pixel resolution of $4096 \times 2160$ and a frame rate of $12\,Hz$. During image acquisition, the water was clear, with the channel bed visible in places. These conditions resulted in a lack of naturally occurring features visible on the water surface that could be used to determine surface velocity. Therefore, throughout the duration of the experiment, spruce wood chips were applied to the water surface from a bridge at the upper extent of the monitored reach. This artificial seeding produced a dense, vivid, and homogeneously distributed pattern of features, the displacement rate of which is considered to equate to the surface velocity. From the video recordings, 1000 images were orthorectified using Photoscan (Agisoft). This was achieved through the input of geographical coordinates relating to 14 GCPs that were visible at varying times throughout image sequence. This approach is discussed further in Detert et al. (2017). The subsequent orthorectified images are presented at a time-step of $0.083\,s$ and the raster pixel scale was consistently set at 64px m$^{-1}$, equivalent to pixel dimensions of $0.0156\,m$

in the x-axis and y-axis. Meta-data describing the scale of the image per pixel, as well as the [x,y] coordinate of the upper left pixel of each image are provided in the corresponding .jgw file.

Reference data was acquired through the deployment of a Teledyne RDI StreamPro ADCP across a single transect in the upper reaches of the studied site. ADCP data was acquired with a bin-depth of $0.02\,\text{m}$ with the upper-most measurement occurring at a distance of $0.14\,\text{m}$ below the water surface. A total of 85 measurements of the velocity magnitude are presented with an average spacing of $0.14\,\text{m}$.

## 2.4 Alpine River, Austria

On the $7^{\text{th}}$ August 2019, aerial surveys were undertaken in order to assess the flow conditions at the turbine outlet of a hydropower dam, the entrance of a fish passage, and the area immediately downstream of these features (Strelnikova et al., 2020). The Alpine river (epithet), is located in Austria, and can be characterised as having a nivo-glacial hydrological regime, with a drainage area of $1057\,\text{km}^2$ and a mean flow discharge of over $32\,\text{m}^3\,\text{s}^{-1}$. At the time of data acquisition the water turbidity was minimal, such that a rocky brown-green riverbed was distinctly visible. Several rocky islands and multiple boulders were located in the middle of the river section of interest. The river section contained turbulent spots and was characterised by heterogeneous flow conditions, with partially opposite flow directions and velocities ranging from 0 to approximately $2\,\text{m}\,\text{s}^{-1}$. Within the study area, the river was up to $35\,\text{m}$ wide, with depths ranging from $0.10$ to $2\,\text{m}$.

Footage of the area was recorded using a DJI Mavic Pro UAS in a hovering mode from an altitude of $50\,\text{m}$ at a frame rate of $25\,\text{Hz}$, with a resolution of $3840 \times 2160\,\text{px}$. The built-in camera of the UAS was directed at nadir. During data acquisition, the flow was artificially seeded with biodegradable cornstarch ecofoam. Individual ecofoam pieces had cylindrical shape, $1.5-2\,\text{cm}$ in diameter and $4.5-6\,\text{cm}$ in length. Tracers were added into the flow from seven locations: over the entrance into the fish ladder, over the turbine outlet, from islands and from both banks. The duration of an acquired video was $5\,\text{min}$. From this video, a dataset of 897 images was extracted at $12.5\,\text{Hz}$ and stabilised using a custom MATLAB script. A subset of the footage was used in a study described by Strelnikova et al. (2020).

For image calibration, eleven GCPs visible in each of the extracted frames were used. Seven GCPs were distributed across both river banks, and four GCPs were located on the islands. The GCPs were surveyed with the use of a differential GPS with an accuracy of $\pm 3\,\text{cm}$. The pixel dimensions of the calibrated imagery are $0.021\,\text{m}$ in the x and y-axis.

An OTT C31 propeller current meter was used to perform reference measurement just below the water surface at 23 locations. During reference data acquisition the propeller axis aligned with the direction of the flow. The duration of measurement at each point was $1\,\text{min}$. The distribution of reference measurements (Figure 3) was selected in a way that described all important components of the heterogeneous flow: the flow from the fish ladder entrance, the dominant flow from the turbine outlet, areas around the main flow curve, and two branches of the main flow after its split. Flow directions were determined using a compass with $10°$ precision in degrees from north. The footage was recorded in a way that north corresponded to $97°$ measured in the clockwise direction from the image top (see the north arrow in the bottom right corner of Figure 3). The locations of reference measurements were determined with the use of a differential GPS with an accuracy of $\pm 3$ cm. The accuracy of the propeller current meter was $\pm 2\%$.

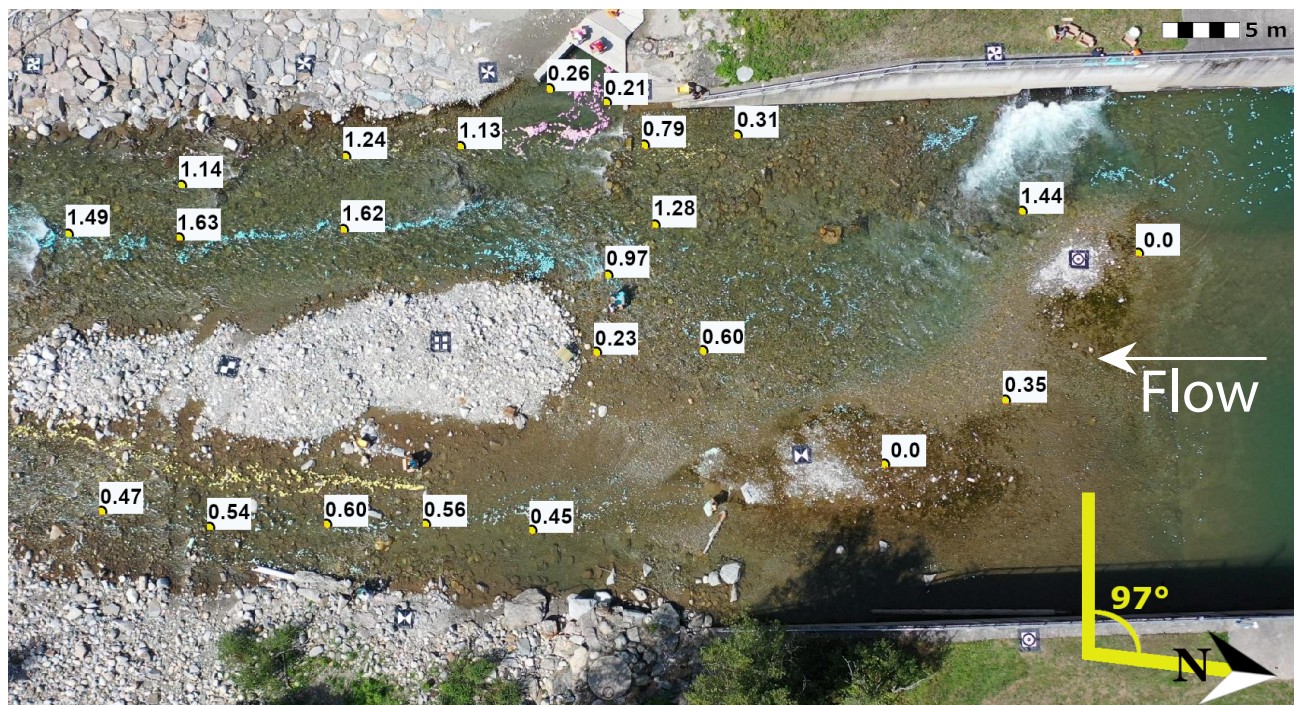

**Figure 3.** A snapshot from the footage collected near a fish ladder and distribution of reference measurements with corresponding velocity magnitudes $(\mathrm{m\,s^{-1}})$. The dominant direction of flow is indicated by the arrow.

## 2.5 River Brenta, Italy

Two distinct experimental approaches have been adopted to generate datasets that describe flows in the $252\,\mathrm{km^2}$ catchment of the River Brenta. The first involved the temporary installation of a GoPro Hero 4 Black Edition camera attached to a telescopic apparatus on the downstream side of a bridge (Tauro et al., 2014). During this deployment, river flow was low with an observed mean velocity of $0.38\,\mathrm{m\,s^{-1}}$. To compensate for the lack of naturally occurring features on the water surface, wood-chips were manually added to the river upstream of the monitoring site resulting in continuous, and relatively homogeneous coverage for the $20\,\mathrm{seconds}$ duration of the image sequences. The camera's field of view was $9.5\times5.3\,\mathrm{m^2}$ and it was configured to collect $1920\times1080$ HD videos at a frame rate of $50\,\mathrm{Hz}$. Distortion of the images as a result of the fish-eye lens was removed using the open-source software *GoPro Studio*. No subsequent orthorectification of the images was required due to the camera apparatus being installed perpendicular to the water surface. The pixel dimensions of the processed imagery were $0.005\,\mathrm{m}$ in the x-axis and y-axis. This could be established either through the projection of two lasers at a fixed and known distance apart to the surface of the river or through identification of a fixed and known object in the field of view. In terms of pre-processing of the imagery, an area of $552\times375$ pixels in the bottom right corner of the images was masked with a black patch. This was to eliminate noise generated by mobile vegetation within the frame. Original RGB images were converted to gray-scale intensity by eliminating hue and saturation information and retaining the luminance. To emphasize lighter particles against a

dark background, images were gamma corrected to darken mid-tones. A total of twelve separate image sequences lasting $20\,\mathrm{s}$, sub-sampled at $25\,\mathrm{Hz}$, and consisting of 500 frames each are presented here.

The second experimental approach involved the temporary deployment of a FLIR Systems AB ThermaCAM SC500. This was suspended from a mobile supporting structure on the downstream side of a bridge at approximately $7\,\mathrm{m}$ above the water surface (Tauro and Grimaldi, 2017). As opposed to capturing images in the usual red, green, and blue bands, this camera is sensitive to thermal infrared radiation, generating a monochrome image with values proportionate to the thermal properties of the objects within the field of view. The application of this approach for image velocimetry requires a distinct thermal signal to be present from either natural (e.g. tributary confluences with water of differing thermal properties), or artificial sources. In this instance, an artificial thermal signal was introduced in the form of ice dices. These were deployed upstream of the bridge and were observed transiting across the field of view as a result of their thermal properties being sufficiently different to that of the water surface. Despite the image resolution being a modest $318 \times 197$ pixels with a frame rate of $5\,\mathrm{Hz}$, this was still sufficient to enable movement of the ice-dices to be tracked. Geometric calibration of the images was achieved by identifying features of known dimensions within the video sequence (i.e. three wooden sticks). The pixel dimensions of the processed imagery are $0.009\,\mathrm{m}$ in the x- and y-axis. Here we present an image sequence consisting of 80 consecutive frames captured over $16\,\mathrm{s}$.

Reference validation data is available in the form of velocity measurements taken at just $3\,\mathrm{cm}$ below the water surface at four locations along the stream cross-section using an OTT Hydromet C2 current meter. At each measurement location 12 replicate measurements were made (Tauro et al., 2017).

## 2.6 La Morge River, France

Within Electricité De France's (EDF) network of over 300 hydrological monitoring stations for the optimal management of water resources, image-based velocimetry approaches have recently been adopted. This approach has been specifically adopted with the aim of reducing uncertainty under high flow conditions (Hauet, 2016). These conditions can develop rapidly, particularly during the summer months as a result of convective storms, posing difficulties for traditional monitoring approaches. However, this setup may also be applied to capture images for the determination of surface velocity under more quiescent conditions. Here, we present images captured on 13[th] January 2015 in the small ($46\,\mathrm{km}^2$), urban catchment of La Morge with a mean altitude of $270\,\mathrm{m}$. Flow conditions were typical with a cross-section width of $7.2\,\mathrm{m}$, mean depth of $0.41\,\mathrm{m}$, and mean velocity of $0.39\,\mathrm{m\,s^{-1}}$. The imaging system used consisted of an analog Panasonic WV-CP500 camera with a focal length of $4\,\mathrm{mm}$. This camera was mounted at an elevated position on a $3\,\mathrm{m}$ pole on the right bank of the channel, oriented in an upstream direction. Images were collected with an effective pixel resolution of $640 \times 480$ at a frame rate of $5\,\mathrm{Hz}$ for a duration of $10\,\mathrm{s}$, resulting in the generation of 48 images. On this occasion, manual seeding of corn chips took place immediately upstream of the camera's field-of-view to enhance the occurrence of features for tracking purposes. This is typically required where natural seeding is inhomogeneous, or completely lacking. Following acquisition of the footage, images were converted to grayscale, and orthorectified using Fudaa-LSPIV to generate images in which 1 pixel represents a real-world distance of $0.01\,\mathrm{m}$.

Reference data was acquired $5\,\mathrm{m}$ downstream of the video acquisition location so as to not interfere with the recorded footage. Therefore, comparisons between measured velocities using image velocimetry and traditional gauging methods in

the same cross-section is not possible. However, a comparison of the computed river discharge from the differing methods is possible. At the upstream location (the camera monitored reach), water depth measurements are available for two transects separated by approximately $6\,\mathrm{m}$ with an average spacing between points of $0.25\,\mathrm{m}$. Through the application of image velocimetry techniques, water depth measurements, and an appropriate value relating the surface velocity to the depth-averaged velocity (estimated to be 0.85), river discharge can be computed. $5\,\mathrm{m}$ downstream of the camera, velocity data was acquired through the use of a mechanical current meter, with measurements taken at 0.2, 0.6, and 0.8 of the river depth. 15 measurements were made along the cross-section, at intervals of $0.5\,\mathrm{m}$. Detailed measurements are provided along with the river discharge computed from these observations. Given the small distance of $5\,\mathrm{m}$ between the location of the recorded video footage and in-stream measurements, and the lack of gains or losses within the reach, river discharge would be the same value at both locations.

## 2.7    St-Julien torrent, France

A high-magnitude flash flood occurred in the St-Julien torrent system during August 2011. This was captured by a local storm chaser using a Canon EOS 5D mark II camera with a $16\,\mathrm{mm}$ fisheye Zenitar lens. Like many headwater systems across Europe, no hydrological monitoring networks are present in this torrent system. Therefore this footage provides a rare insight into the hydraulic processes occurring during a flash flood in a steep, small ($20\,\mathrm{km^2}$), torrent system where mean flow velocities are approximately $6\,\mathrm{m\,s^{-1}}$. The footage itself was recorded at a resolution of $1920 \times 1080\mathrm{px}$ at a frame rate of $25\,\mathrm{Hz}$ (Figure 4a). The footage was not filmed from a fixed location therefore complications involving camera movement, and orthorectification had to be overcome. These steps are explained in detail in Le Boursicaud et al. (2016). Following correction for these factors, a sequence of 51 consecutive and geometrically stable images are produced (Figure 4b). Each pixel width represents a metric scale of $0.03\,\mathrm{m}$. Despite the lack of detailed reference velocity measurements for this case study, researchers interested in reconstructing flash flood processes may find it valuable to understand how the range of available methods perform relative to each other given that image velocimetry techniques perhaps offer the best opportunity to estimate flows under these extreme conditions.

## 2.8    River La Vence, France

On May 8th, 2019, a Samsung Galaxy S7 was utilised to capture footage for image velocimetry analysis on the River La Vence, a $63.75\,\mathrm{km^2}$ catchment in France. At the time of deployment, the river width was approximately $6.3\,\mathrm{m}$, with a river stage of $0.44\,\mathrm{m}$. A discharge of $1.15\,\mathrm{m^3\,s^{-1}}$, and mean velocity of $0.65\,\mathrm{m\,s^{-1}}$ were observed. The camera was fixed in location with the camera lens angled at approximately $31°$ from nadir and the image x-axis at approximately perpendicular to the direction of flow. Images were collected for a duration of $5\,\mathrm{s}$ at a resolution of $1920 \times 1080\mathrm{px}$ and frame rate of $30\,\mathrm{Hz}$. The presence of visible turbulent structures advecting downstream offer the potential for the extraction of surface velocity information from this footage. Image pre-processing consisted of orthorectification, and color conversion to gray-scale. 150 consecutive images which have been subjected to these processing steps are presented here at their original frame rate of $30\,\mathrm{Hz}$. The pixel dimensions of the processed imagery are $0.008\,\mathrm{m}$ in the x and y-axis.

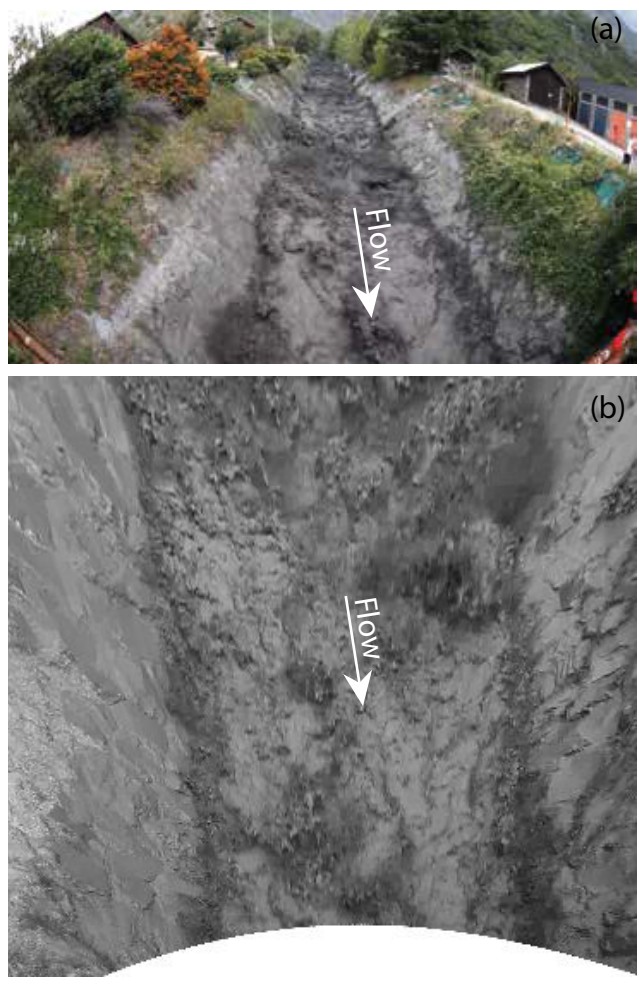

**Figure 4.** (a) Original footage of a flash flood in the St-Julien torrent acquired by a storm chaser equipped with Canon EOS 5D mark II camera; (b) Orthorectified and geometrically stable image with the field of view clipped to the lower 50% of the image. The direction of flow is indicated by the arrows.

Reference data was acquired by means of a HydroProfiler M-pro ADCP and consists of a single transect consisting of 8 measurements with an average spacing of $0.7\,\mathrm{m}$. ADCP data was acquired with a bin-depth of $0.003\,\mathrm{m}$ with the upper-most measurement occurring at a distance of $0.101\,\mathrm{m}$ below the water surface.

## 2.9 River Tiber, Italy

A permanent gauge station on the River Tiber, Italy was installed to test the feasibility for automated image velocimetry methods to quantify the flow rates of a major European river with a catchment area of $17\,460\,\mathrm{km}^2$. This deployment involves the use of a Mobotix FlexMount S15 IP camera attached to the underside of Ponte del Foro Italico, in the city of Rome (Tauro et al., 2016a). The wide angle lens on the SP15 camera introduces distortion into the images, which was subsequently removed using the Adobe Photoshop Lens correction filter. In a similar setup to the first of the River Brenta approaches, this camera

is positioned orthogonal to the water surface, thereby circumventing the need for orthorectification of the generated images. Transformation of the camera pixels (px) to metric units (m) was again achieved by firing lasers of a known distance apart at the water surface. The image can be scaled to metric distances given: $1\mathrm{px} = 0.016\,\mathrm{m}$. The camera itself generated videos with a resolution of $2048 \times 768\mathrm{px}$. However these were sub-sampled to $865 \times 530\mathrm{px}$ at a frame rate of $\approx 6.95\mathrm{Hz}$ during pre-possessing. The data specifically presented here consists of 410 consecutive frames collected over a $60\,\mathrm{second}$ period during a moderate

flood event in February 2015. At the time of acquisition, the river stage was $7.23\,\mathrm{m}$, and average surface velocity (measured by a RVM20 speed surface velocity radar) was $2.33\mathrm{m\,s}^{-1}$ (Tauro et al., 2017). Whilst only a single reference velocity value is available, this measurement is representative of the surface velocities within a surrounding area of approximately $3 \times 3\,\mathrm{m}^2$. The approximate spatial footprint of the surface velocity radar measurement is provided in pixel units.

## 2.10 River Bradano, Italy

On $14^{\mathrm{th}}$ October 2016, an experiment was undertaken in order to explore the optimal setup for the acquisition of surface flow velocity measurements using an UAS (Dal Sasso et al., 2018). The experiment took place in the valley portion of the Bradano River, located in the Basilicata region of Italy. This large alluvial river has a catchment area of $2581\,\mathrm{km}^2$ and is characterised by low gradient (0.1%) and low relative submergence (Dal Sasso et al., 2018). At the time of the experiment, the cross-section width was $11.4\,\mathrm{m}$, with a maximum depth of $0.80\,\mathrm{m}$. The average surface velocity was $0.75\mathrm{m\,s}^{-1}$ and total discharge was

$3.97\,\mathrm{m}^3\,\mathrm{s}^{-1}$. During the experiment, a DJI Phantom 3 Pro UAS equipped with a Sony EXMOR 1/2.3" CMOS camera sensor was deployed.

The UAS hovered over the centre of the River Bradano with a nadir camera positioned perpendicular to the direction of flow. An area of $17.0 \times 9.6\mathrm{m}^2$ was imaged, including the entire cross-section of interest (with a width of approximately $11.4\,\mathrm{m}$). Video footage was captured for a duration of $1\,\mathrm{min}\ 43\,\mathrm{s}$ at a pixel resolution of $1920 \times 1080$, and a frame rate of $24\,\mathrm{Hz}$.

Due to the high turbidity of the flow, there is a weak natural contrast across the image which diminishes the number of naturally occurring, visible tracers. Therefore, throughout the duration of the footage, operatives manually introduced charcoal to the water surface immediately upstream of the monitoring site. The color of these particles was sufficiently distinct from the background to enable their displacement to be optically tracked. However, the distribution of these tracers is generally

limited to the central portion of the flow which may limit the availability of traceable features towards the channel boundaries. Following collection of the footage, a number of processing steps were subsequently undertaken. This included conversion of the grayscale images to black and white, and contrast correction in order to more prominently highlight the artificial tracers on the water surface. 600 images which have been subjected to these processing steps are available at their original resolution and frame rate. An addition processing step involved the stabilisation of the image sequence to minimise apparent movement of the platform. Transformation of the images from pixel units to metric distance can be achieved using the following function: 1px = $0.009\,\mathrm{m}$. Validation data in the form of surface velocities was obtained at seven points in the cross-section, at $1\,\mathrm{m}$ intervals using a Seba F1 current flow meter. The locations of these measurements are provided in pixels relative to the first frame of the stabilised image sequence.

## 2.11    River Noce, Italy

On 26[th] July 2017, in the middle reaches of the $413\,\mathrm{km}^2$, single-thread, River Noce, a DJI Phantom 3 Pro UAS Sony EXMOR 1/2.3" CMOS sensor was deployed to capture footage for image velocimetry analysis (Dal Sasso et al., 2018). At the time of deployment, water levels were low with an observed discharge of $1.70\,\mathrm{m}^3\,\mathrm{s}^{-1}$ and mean velocity of $0.43\,\mathrm{m\,s}^{-1}$. Turbidity was also minimal resulting in the gravel bed being distinctly visible in the footage. The camera was oriented with its x-axis perpendicular to the water surface enabling the $14.6\,\mathrm{m}$ wide channel to be fully observed (Figure 5a). Images were collected for a duration of $1\,\mathrm{min}\,48\,\mathrm{s}$ at a resolution of $3840 \times 2160\mathrm{px}$ and frame rate of $24\,\mathrm{Hz}$. The clear water and bright sunlight results in non-homogeneous illumination of the water surface. This is particularly apparent in the lower left quarter of the video frame. Naturally occurring tracers are also largely absent making these challenging conditions for the application of image velocimetry techniques. To offset these issues, wood chips were introduced upstream of the monitoring location. These features were visibly brighter than the background enabling their transition to be detected optically. Image processing consisted of contrast stretching and conversion of grayscale images to black and white in order to enhance the visibility of the artificial tracers against the background (Figure 5b). 70 consecutive images which have been subjected to these processing steps are presented here at a downscaled resolution of $1920 \times 1080\mathrm{px}$ and frame rate of $12\,\mathrm{Hz}$. Following sub-sampling, each pixel in the image represents a distance of $0.009\,\mathrm{m}$ in metric units. Validation data in the form of surface velocities were obtained at thirteen locations, at $1\,\mathrm{m}$ intervals, along the cross-section using a Seba F1 current flow meter. The locations for each of these measurements is provided in pixel units.

## 2.12    Castor River, Canada

Here we present footage acquired from the middle reaches of the Castor River in Ontario, Canada ($45.26194°$ Latitude, -$75.34444°$ Longitude). At this location, the channel is a stable, single thread, meandering river with a catchment contributing area of $439\,\mathrm{km}^2$. Footage was acquired on two separate occasions, consisting of very different flow conditions:

The first set of videos were acquired on 10[th] April 2019 using a Hikvision DS-2CD2T42WD-I5 4mm IP camera. This was mounted on the left bank at an oblique angle of $57°$ from nadir. Video footage was captured consisting of three, $30\,\mathrm{second}$ videos at a resolution of $2688 \times 1520\mathrm{px}$ and frame rate of $20\,\mathrm{Hz}$. The first $2-3\,\mathrm{s}$ of each recording have been removed from the

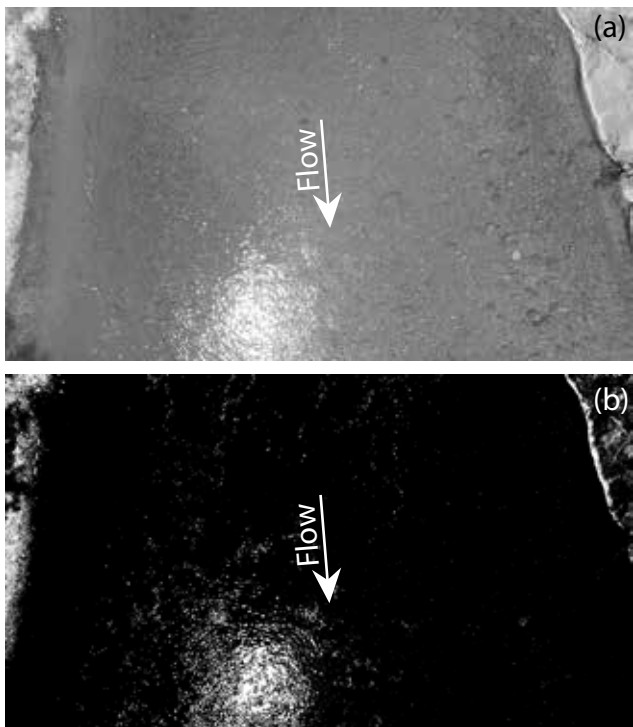

**Figure 5.** (a) Grayscale footage acquired by the Phantom 3 UAS over the River Noce, and (b) following contrast stretching. The direction of flow is indicated by the arrows.

submission as these frames experienced compression and frame rate issues. However, the remainder of the video is unaffected. The videos were captured over a duration of approximately $4.5\,\text{h}$ and over this time the river stage was stable, varying between $3.772\,\text{m}$ at 11:25, $3.769\,\text{m}$ at 13:45, and $3.77\,\text{m}$ at 15:55 (local time). Under these moderate flow conditions, mean velocity was observed to be $1.26\,\text{m}\,\text{s}^{-1}$, with mean and maximum depths of $0.80$ and $1.19\,\text{m}$ respectively within the $27\,\text{m}$ wide river.

5   No image stabilisation was performed on the image sequence and the imagery was orthorectified using KLT-IV and the use of twelve ground control points. These control points were placed at varying heights across both sides of the channel, and the distances, horizontal and vertical angles between points were surveyed using a tripod mounted Leica S910. This enabled a local coordinate system to developed relative to a local benchmark. In the resultant imagery each image pixel represents a distance of $0.01\,\text{m}$ in metric units.

10   Reference data was acquired through the deployment of a Teledyne RDI StreamPro ADCP with four transects being completed across a single cross-section. ADCP data was acquired with a bin-depth of $0.05\,\text{m}$ with the upper-most measurement occurring at a distance of $0.17\,\text{m}$ below the water surface. Between 149 and 219 velocity magnitude measurements are reported for each transect with an average spacing between measurements of $0.12 - 0.18\,\text{m}$. The location of each velocity magnitude measurement is reported in pixel units based on the orthorectified imagery.

The second video set obtained at Castor River was acquired on 9[th] July 2019 and consists of a single, $27\,\mathrm{s}$ video. This was acquired from the left bank using a ACTI A31 IP camera, mounted at an oblique angle of $54°$ from nadir. Video footage was recorded at a resolution of $1920 \times 1080\mathrm{px}$ and frame rate of $30\,\mathrm{Hz}$. At the time of acquisition, river levels were low, with a reported stage of $3.128\,\mathrm{m}$. At this time, the river was $21\,\mathrm{m}$ wide with a mean and maximum depth of $0.45$ and $0.62\,\mathrm{m}$
respectively. Observed discharge was $0.926\,\mathrm{m^3\,s^{-1}}$ with a mean velocity of $0.13\,\mathrm{m\,s^{-1}}$. No image stabilisation was performed on the image sequence and the imagery was orthorectified using KLT-IV and the same ground control points as the previous set of videos. In the resultant imagery each image pixel represents a distance of $0.01\,\mathrm{m}$ in metric units. Reference data was acquired using a FlowTracker2 handheld acoustic Doppler velocimeter. Velocity measurements were made at four locations along a single cross-section and at percentage depths of 0 (i.e. water surface), 20, 40, 60, 80, and 100%. The x and y velocity
components are reported along with the mean velocity. The location of each velocity measurement is reported in pixel units based on the orthorectified imagery.

## 2.13  Salmon River, Canada

On the 4[th] June 2019, a DJI Phantom 4 Pro was used to acquire footage over the Salmon River in British Columbia, Canada ($50.312222°$ Latitude, $-125.907500°$ Longitude). Footage was acquired immediately downstream of the confluence between
the Salmon River and the smaller White River. Here, the catchment contributing area is $1210\,\mathrm{km^2}$, and a $59\,\mathrm{m}$ wide, single thread channel is present. At the time of image acquisition, river levels were low, with an average depth of $0.65\,\mathrm{m}$, a reported discharge of $22.9\,\mathrm{m^3\,s^{-1}}$, and mean velocity of $0.65\,\mathrm{m\,s^{-1}}$. A $1\,\mathrm{min}$ video was collected with a view angle of approximately nadir whilst hovering at an elevation of $102\,\mathrm{m}$ over the field of interest. The footage was acquired at a resolution of $1920 \times 1080\mathrm{px}$ and a frame rate of $24\,\mathrm{Hz}$. Present within the field of view are four ground control points, located on both sides of the
channel. The straight-line distances between each of the ground control points were measured and a local coordinate system developed following the principles of trilateration. A two-stage processing method was adopted to generate imagery suitable for velocimetry analysis. This consisted of: (i) image stabilisation; and (ii) orthorectification. These were performed using the built-in functionality of KLT-IV (Table A1). Following processing, each image pixel represents a distance of $0.01\,\mathrm{m}$ in metric units. Reference velocity data was acquired using a FlowTracker handheld acoustic Doppler velocimeter and this consists of
measurements at twenty six locations along a single cross-section at intervals of approximately $3\,\mathrm{m}$. These measurements were obtained at 60% of the water depth and the mean velocity is reported. The location of each velocity measurement is reported in pixel units based on the orthorectified imagery.

## 3  Conclusions

Applied hydrology research, focusing on the quantification of fluid flow processes in river systems, has been greatly enhanced
by the availability of large-scale image velocimetry techniques (e.g. Table A1). The flexibility of these approaches has led to improvements in the understanding of hydrological processes in otherwise difficult to access environments. This has been possible through image capture using a range of platforms including: unmanned aerial systems, thermal infra-red cameras,

Go-Pro's, and IP cameras, which enable non-contact sensing of the waterbody. Consequently, a growing, but disparate, range of imagery datasets have been produced (e.g. Table A2). Here we collate and describe a range of these example datasets, most of which have validation data in the form of velocity measurements undertaken using standard operational approaches (e.g. current flow meter, ADCP, radar).

5     This unique dataset represents the first step in creation of a community database for image velocimetry benchmarking studies. It offers the hydrological community the opportunity to assess the accuracy of existing approaches under a range of conditions. Key comparisons may be made surrounding the relative impact of the seeding characteristics (e.g. River Arrow, Murg River), the type of sensor used (e.g. River Brenta), the potential for background noise, e.g. glare, visible river bed, to be filtered (e.g. Salmon River), and the impacts of stabilisation on velocity outputs (e.g. River Bradano, Alpine River).

10     The generation of similar datasets of images are widely used to evaluate the effectiveness and accuracy of algorithms in related fields such as fluid mechanics (e.g. Okamoto et al., 2000), and we envisage such a dataset for large-scale fluvial environments will encourage further scientific assessment and development of image velocimetry approaches.

    Though the diversity of experimental settings and data formats included in this dataset may limit the inter-comparability of experiments, this dataset is well suited for comparison of different algorithms within the framework of one selected study. An 15 advantage of the dataset is that it is representative of the multitude of possible experimental settings. Techniques tested and tailored with the help of such a diverse dataset are expected to be more robust, and their limitations are expected to be easier to identify. Ultimately, forensic assessment of these techniques will provide researchers and competent authorities with a greater understanding of their applicability. Further efforts will be put into extending the dataset and unifying data formats both for optical data and ground truth data included, with a goal to create a standardised database which explicitly facilitates testing of 20 a selected technique in different experimental settings.

## 4   Data availability

Datasets presented in this manuscript can be readily downloaded from the following website: http://doi.org/10.4121/uuid: 014d56f7-06dd-49ad-a48c-2282ab10428e. Data includes the footage/imagery required for image velocimetry analysis, plus validation data for 12 of the 13 case studies presented. Please contact the corresponding author if further details are required 25 (Perks et al., 2020).

**Table A1.** Details of software developed for image velocimetry analysis

| Software | Key Functions | Availability |
|---|---|---|
| Fudaa-LSPIV[a] | Sample images from movies, image orthorectification, cross-correlation, data filtering, discharge computation | Open source interface, free executables |
| KLT-IV[b] | Lens distortion removal, image stabilisation and orthorectification, tracking individual trajectories, discharge computation | Proprietary software |
| KU-STIV[c] | Distortion removal, orthorectification, image stabilisation, image pattern coherence | Proprietary software |
| LSPIV app[d] | Camera calibration, image orthorectification, cross-correlation, image pattern coherence | Free app for Android and iOS |
| MAT PIV[e] | Image coordinate transformation, cross-correlation, post-processing filters | Free toolbox for MATLAB |
| OTV[f] | Tracking individual trajectories and average surface flow velocity estimation | Proprietary software |
| Photrack. SSIV[g] | Image orthorectification, cross-correlation, flow surface structure filtering, results filtering, discharge estimation. Stand-alone camera system for continuous measurement (DischargeKeeper), or in a smart-phone application (DischargeApp) | Proprietary software |
| PIVlab[h] | Image pre-processing, direct cross-correlation, discrete Fourier transform, sub-pixel solutiona, post-processing tools | Free toolbox for MATLAB |
| PTVlab[i] | Image pre-processing, cross-correlation, relaxation algorithm, dynamic threshold binarization, iterative relaxation, tracking of individual trajectories, post-processing tools | Free toolbox for MATLAB |
| PTV-Stream[j] | Tracking individual trajectories and average surface flow velocity estimation | Proprietary software |
| RIVeR[k] | Image extraction from video, image processing (PIVlab or PTVlab), rectification of velocities to real-world units, discharge calculation | Free toolbox for MATLAB |

[a]Le Coz et al. (2014); [b]Perks et al. (2016); [c]Fujita et al. (2007); [d]Tsubaki (2018); [e]Sveen and Cowen (2004); [f]Tauro et al. (2018b); [g]Leitão et al. (2018); [h]Thielicke and Stamhuis (2014); [i]Brevis et al. (2011); [j]Tauro et al. (2019); [k]Patalano et al. (2017)

**Table A2.** Experimental setup during image acquisition, details of subsequent image pre-processing, availability of validation data and published analysis.

| Identifier | Image Acquisition | Pre-processing | Validation Data | Image Velocimetry Software Used | Published Analysis |
|---|---|---|---|---|---|
| River Arrow (a) | DJI Phantom Pro 4 UAS | Conversion to grayscale intensity Orthorectification Image sequence sub-sampled | Five cross-sections of 9-10 points using a Valeport ECM | Fudaa-LSPIV | N/A |
| River Arrow (b) | Go Pro Hero 4 | As above | See Arrow (a) | Fudaa-LSPIV | N/A |
| River Thalhofen | Vivotek IB836BA-HT | Orthorectification Conversion to grayscale intensity | A single RiverPro ADCP transect | Photrack. SSIV | N/A |
| Murg River | DJI Phantom FC40 UAS with GoPro Hero3+ | Orthorectification | A single StreamPro ADCP transect | PIVlab | Detert et al. (2017) |
| Alpine River | DJI Mavic Pro with Hasselblad 1/2.3" CMOS sensor | None | Water surface velocities measured using an OTT C31 at 23 locations across the field of view | PIVlab | Strelnikova et al. (2020) |
| River Brenta (a) | GoPro Hero 4 | Distortion removal Gamma correction | Velocity measurements 3 cm below water surface at four locations in a single cross-section using an OTT C2 | PIVlab & PTVlab | Tauro et al. (2017) |
| River Brenta (b) | FLIR SC500 | Orthorectification Extraction of RGB from thermal | See Brenta (a) | PTVlab | Tauro and Grimaldi (2017) |

| Identifier | Image Acquisition | Pre-processing | Validation Data | Image Velocimetry Software Used | Published Analysis |
|---|---|---|---|---|---|
| La Morge | WV-CP500 | Orthorectification | 15 paired velocity and depth measurements performed 5 m downstream of camera, and depth across two transects within camera field of view | Fudaa-LSPIV | Hauet (2016) |
| St-Julien torrent | Canon EOS 5D | Distortion removal Orthorectification Image stabilisation | N/A | Fudaa-LSPIV | Le Boursicaud et al. (2016) |
| River La Vence | Samsung Galaxy S7 | Orthorectification Conversion to grayscale intensity | A single HydroProfiler M-pro ADCP transect | Photrack. SSIV | N/A |
| River Tiber | Mobotix S15 | Distortion removal Conversion to grayscale intensity | A single RVM20 SVR measurement | PIVlab & PTVlab | Tauro et al. (2017) |
| River Bradano | DJI Phantom 3 Pro UAS with Sony 1/2.3" CMOS sensor | Conversion to black and white images Contrast correction | Surface velocities at 7 points within a single cross-section using a SEBA F1 | PTVlab | Dal Sasso et al. (2018) |
| River Noce | DJI Phantom 3 Pro UAS with Sony 1/2.3" CMOS sensor | Contrast stretching Conversion to black and white images Image sequence sub-sampled | Surface velocities at 13 points within a single cross-section using a SEBA F1 | PTVlab | Dal Sasso et al. (2018) |
| Castor River (a) | Hikvision DS-2CD2T42WD-I5 4mm IP camera | Conversion to grayscale Orthorectified | Four StreamPro ADCP transects at a single cross-section | KLT-IV | N/A |
| Castor River (b) | ACTI A31 IP camera | Conversion to grayscale Orthorectified | Velocity measurements at four points along a single cross-section at six depths using a FlowTracker2 ADV | KLT-IV | N/A |

| Identifier | Image Acquisition | Pre-processing | Validation Data | Image Velocimetry Software Used | Published Analysis |
|---|---|---|---|---|---|
| Salmon River | DJI Phantom 4 Pro | Conversion to grayscale Stabilised Orthorectified | Velocity measurements at 24 points in a single cross-section using a FlowTracker ADV | KLT-IV | N/A |

*Author contributions.* This article and dataset compilation was initially proposed by SM and SG. MP led the production of the manuscript. Each co-author contributed to the writing, and the contribution of datasets.

*Competing interests.* The authors declare that they have no conflict of interest.

*Acknowledgements.* This work was funded by the COST Action CA16219 "HARMONIOUS—Harmonization of UAS techniques for agri-
cultural and natural ecosystems monitoring". SG and FT acknowledge support by Ministero degli Affari Esteri project 2015 Italy-USA
PGR00175, by POR-FESR 2014-2020 n. 737616 INFRASAFE, and by "Departments of Excellence-2018" Program (Dipartimenti di Ec-
cellenza) of the Italian Ministry of Education, University and Research, DIBAF-Department of University of Tuscia, Project "Landscape
4.0 – food, wellbeing and environment". The contribution of the Murg River dataset was kindly made available by Dr Martin Detert.
ADCP measurements for this case study were provided by Urs Vogel (Limnex AG). The Murg River dataset can also be accessed at:
https://figshare.com/articles/S1S2S3_Murg_20160406_zip/4680715/1. The authors wish to thank Georgy Ayzel, two anonymous reviewers,
and the handling Editor for their detailed comments, which greatly improved the quality of this manuscript.

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
