# Peer review of "Towards harmonization of image velocimetry techniques for river surface velocity observations"

_Earth System Science Data, 2019_

## Referee Comment (RC1) · Anonymous Referee #1 · 10 Oct 2019

This contribution is unique and important for the society of research and development on the image-based hydrometry approach. Provided images are useful for the RD mentioned above. Other than images, information for validation is described in the manuscript. The unfortunate thing is that quantitative validation data was not included in the provided data set. This restricts the aim of this contribution, "validation and accuracy assessment" (the last sentence in an abstract.) To accomplish the objective of the study further, I suggest some modifications in both the manuscript and the data-set.

Point to point comments

Lines 11-14, page 2: Two sentences are discussing the image velocimetry application in labs and the logical flow between two sentences is difficult to follow. To make this

[Figure]

part easier for reading, one option is to move "wide variety of experimental conditions" at the beginning part of the second sentence, since this part is a distinguishing point to the first sentence.

Figure 1. I suggest dropping "Geographical" from the caption or add some more information regarding geography in the figure, e.g. water network, river basin, elevation etc.

Figures 2 and 3. Original and rectified images are provided in each figure and I guess the directions are rotated. Better to indicate the direction of the flow, e.g. by putting the arrow with a label of "flow" onto each panel.

Table A2. Label, this is quite a minor thing but I suggest use "Image Aquisition" instead of "Data Aquisition" in the label.

Table A2. Validation data, I suggest to add the description about validation data (e.g. how and where).

Table A2. Flow information, I suggest adding the mean velocity, representative depth, Froude number, width etc. (maybe, rotate 90 degrees the table to expand the width of the table).

Data-set. Better to include movie file for each site for making easier to know the image characteristics and image recording approaches. (I made by myself for the purpose of review, and I can share it if needed.) Also suggested is providing text file(s) specifying the image resolution, location of the edges of images, and frame rate, and/or provide e.g. jgw, tfw and pgw files for corresponding image/folder (for jpg, tiff and png image, respectively).

Data-set. For sites with velocity distribution measured for validation, provide the location and velocity of the data as e.g. CSV file.

Data-set. Type of image file and the structure of file name differ for each folder, this making the pre-processing a bit troublesome for a potential user of the data. Could you

provide also a unified formatted image set, e.g. 0000.png? (I made this also by myself for review, and I can share it if needed.)

I understand my suggestions are messy but be making a better contribution of this work to the RD society in this field.
* * *

---

## Referee Comment (RC2) · Georgy Ayzel (Referee) · 5 Nov 2019

General comments

The present manuscript is aimed to introduce the new dataset, which will help to systematize and benchmark the emerging techniques for image-based river surface velocity estimation. The corresponding dataset consists of pre-processed videos from 12 research sites located in six different countries and covered a wide range of fluvial settings.

In my opinion, the introduced dataset has sound potential and of high interest in the research community. However, I recommend authors to provide major revisions which may help to increase the dataset value for the target community and make it the

first benchmark dataset for image-based velocimetry techniques (e.g., as the MNIST database for image classification).

Specific comments

1. Abstract (Page 1, Ln 10): It is mentioned that 13 case studies have been presented in the dataset, but Section 2 describes only 12.

2. Section 2.7 St-Julien torrent, France (Page 8, Ln 24-31): As for this particular case study, the validation data is unavailable, the explicit description is needed to clarify the reasons behind the inclusion of the corresponding data to the introduced dataset. At least, it is not clear how this data will help to pursue one of the dataset objectives as "testing specific image velocimetry techniques."

3. Section 2.9 River Tiber, Italy (Page 10, Ln 13-24): In my opinion, the single measurement of average velocity, which is provided as validation data for this site has limited value for the comprehensive analysis of different image velocimetry techniques reliability and efficiency. Please, provide explicit reasoning why this data will also help to meet the declared dataset objectives.

4. Dataset: I have realized that for some sites (e.g., Arrow River, Bradano River), scenes are not aligned with each other, i.e., ground (river banks) is not stable. In my opinion, key point alignment is needed to simplify the use of the dataset. This way, if the ground is stable for all the scenes, optical flow techniques can be easily implemented out-of-the-box for velocity field estimation.

5. Dataset: I recommend authors to consider the change of format for the provided images to GeoTIFF (or similar) to provide explicit georeferencing capabilities. It will substantially simplify the validation procedure by providing a solid basis for validation data georeferencing.

6. Dataset: I did not find any validation data mentioned in the manuscript (Section 2) in the provided dataset archive.

7. Dataset: In my opinion, the additional section which will confirm the introduced dataset validity, and its corresponding value for the target community is needed. The potential reader has to be sure that the dataset is consistent with the declared objectives and therefore serves the reader's needs the best (e.g., benchmarking the new technique/software). I recommend authors to provide a brief analysis of the single case study showing the extracted velocities and comparing them to the validation data. Authors also may consider supporting the corresponding analysis with a code example – this may significantly increase the reader's interest to the dataset and manuscript itself.

---

## Referee Comment (RC3) · Anonymous Referee #3 · 7 Nov 2019

This paper compiled river surface velocity observations using image velocimetry techniques over eleven sites across Europe and one site in India. It describes method used in the data acquisition, pre-processing software/functions, and the hydro-geomorphic setting at each site to generate velocity measurement. It is exciting to see the new approach for determining flow velocity that can be executed even with a smart phone (Samsung Galaxy S7). I think that the work is valuable and interested in the hydrology community for the development of image-based techniques, which could be further applied in modeling and monitoring. However, it is not clear to me what contributions this paper offers. The abstract mentions inter-comparison and validation of the various techniques, but they were not actually performed, which seems to be missing a major component of the paper. The validation data exists for most cases, then why

not present the resulting datasets in the form that is directly compared and validate, instead of the image clips? I am having a hard time grasping how the results of this paper can be used as benchmark datasets in the current form. Even if quantitative validation is addressed, the measurements are taken at specific time and location of the river (i.e. specific hydro-geomorphic setting), so it may not be comparable if someone uses different camera and processing technique at different time and/or location. I understand that the nature of the observation and approach is not suitable for generalization, but the paper in the current form doesn't seem to fit into the context of "towards harmonization of the techniques". Therefore, I recommend major revision at this time.

---

## Author Comment (AC1) · 9 Jan 2020

**Reviewer 1 Comments (RC1)**

**Reviewers Comment (RC) 1.1:** This contribution is unique and important for the society of research and development on the image-based hydrometry approach. Provided images are useful for the RD mentioned above. Other than images, information for validation is described in the manuscript. The unfortunate thing is that quantitative validation data was not included in the provided data set. This restricts the aim of this contribution, "validation and accuracy assessment" (the last sentence in an abstract.) To accomplish the objective of the study further, I suggest some modifications in both the manuscript and the dataset.

**Authors Comment (AC) 1.1:** Thanks for the considered review of our submitted article. In response to comments from Reviewers 1, 2 and 3, the revised version of the manuscript will fully describe the validation data available for each of the case studies. This validation data will be collated, standardised and published in the data archive along with the orthorectified images, which were presented in the original submission.

**RC 1.2:** Lines 11-14, page 2: Two sentences are discussing the image velocimetry application in labs and the logical flow between two sentences is difficult to follow. To make this part easier for reading, one option is to move "wide variety of experimental conditions" at the beginning part of the second sentence, since this part is a distinguishing point to the first sentence

**AC 1.2:** The sentences in question will be revised to read: 'Image velocimetry involves the application of cross-correlation, or computer vision techniques to a series of consecutive images (or extracted video frames) to generate vectors of water velocities across a field-of-view. Despite being originally developed for use in highly controlled laboratory settings (e.g., Dudderar and Simpkins, 1977; Adrian, 1984; Pickering and Halliwell, 1984), it has since been applied to a wide variety of experimental conditions.'

**RC 1.3:** Figure 1. I suggest dropping "Geographical" from the caption or add some more information regarding geography in the figure, e.g. water network, river basin, elevation, etc.

**AC 1.3:** The Figure caption will be modified to read: 'Locations of the monitoring sites from which data are presented: (a) River Arrow, UK; (b) River Dart, UK; (c) River Thalhofen, Germany; (d) Murg River, Switzerland; (e) River Brenta, Italy; (f) La Morge, France; (g) St-Julien torrent, France; (h) River La Vence, France; (i) River Tiber, Italy; (j) River Bradano, Italy; (k) River Noce, Italy. Not shown: River Karehalla, India. Map spatial reference: ETRS (1989).'

**RC 1.4:** Figures 2 and 3. Original and rectified images are provided in each figure and I guess the directions are rotated. Better to indicate the direction of the flow, e.g. by putting the arrow with a label of "flow" onto each panel.

**AC 1.4:** For both Figures 2 and 3 the flow direction will be indicated using an arrow. The Figure 2 label will also be modified to describe the flow direction.

**RC 1.5:** Table A2. Label, this is quite a minor thing but I suggest use "Image Aquisition" instead of "Data Aquisition" in the label.

**AC 1.5:** The label of Table A2 will be modified to read: 'Image Acquisition'.

**RC 1.6:** Table A2. Validation data, I suggest to add the description about validation data (e.g. how and where).

**AC 1.6:** In order to support the presentation of the validation data we will use this part of the table to provide a summary of the reference measurements undertaken for each case-study (e.g. instrument, number of points, duration of sample).

**RC 1.7:** Table A2. Flow information, I suggest adding the mean velocity, representative depth, Froude number, width etc. (maybe, rotate 90 degrees the table to expand the width of the table).
**AC 1.7:** Unfortunately, this information is not readily available for all of the case-studies. However, we will ensure that all relevant information describing the hydrological conditions for each case-study is presented within each sub-section of the text.

**RC 1.8:** Data-set. Better to include movie file for each site for making easier to know the image characteristics and image recording approaches. (I made by myself for the purpose of review, and I can share it if needed.) Also suggested is providing text file(s) specifying the image resolution, location of the edges of images, and frame rate, and/or provide e.g. jgw, tfw and pgw files for corresponding image/folder (for jpg, tiff and png image, respectively).
**AC 1.8:** For each case-study a video will be created using the orthorectified images. This video will be produced at the same image resolution as the orthorectified images, and at the frame rate required for analysis. Unfortunately, the geographical coordinates are unknown for many of the image sequences presented. In the majority of cases, the ground control points were surveyed using an instrument that utilises a local reference (e.g. total station). In these cases, it would be inappropriate to provide a tgw, tfw, etc. file.

**RC 1.9:** Data-set. For sites with velocity distribution measured for validation, provide the location and velocity of the data as e.g. CSV file.
**AC 1.9:** In response to comments from Reviewers 1, 2 and 3, the revised version of the manuscript will fully describe the validation data available for each of the case studies. This validation data will be collated, standardised and published in the data archive along with the orthorectified images, which were presented in the original submission. We will provide this reference data in a .csv format.

**RC 1.10:** Data-set. Type of image file and the structure of file name differ for each folder, this making the pre-processing a bit troublesome for a potential user of the data. Could you provide also a unified formatted image set, e.g. 0000.png? (I made this also by myself for review, and I can share it if needed.)
**AC 1.10:** The filenames will be altered to have a standardised format e.g. 0000.png.

**Reviewer 2 Comments (RC2)**

**RC 2.1:** The present manuscript is aimed to introduce the new dataset, which will help to systematize and benchmark the emerging techniques for image-based river surface velocity estimation. The corresponding dataset consists of pre-processed videos from 12 research sites located in six different countries and covered a wide range of fluvial settings. In my opinion, the introduced dataset has sound potential and of high interest in the research community. However, I recommend authors to provide major revisions which may help to increase the dataset value for the target community and make it the first benchmark dataset for image-based velocimetry techniques (e.g., as the MNIST database for image classification).
**AC 2.1:** We would like to thank the reviewer for taking the time to provide a thorough review of our submitted manuscript.

**RC 2.2:** Abstract (Page 1, Ln 10): It is mentioned that 13 case studies have been presented in the dataset, but Section 2 describes only 12.
**AC 2.2:** This is a typographical error and will be corrected. The abstract should read: 'Validation data is available for 11 of the 12 case studies presented enabling these data to be used for validation and accuracy assessment'.

**RC 2.3:** Section 2.7 St-Julien torrent, France (Page 8, Ln 24-31): As for this particular case study, the validation data is unavailable, the explicit description is needed to clarify the reasons behind the inclusion of the corresponding data to the introduced dataset. At least, it is not clear how this data will help to pursue one of the dataset objectives as "testing specific image velocimetry techniques."
**AC 2.3:** This particular case study represents a flash flood, which occurred in a torrent system in France producing mean velocities of approx. 6 m s$^{-1}$. Whilst no detailed reference measurements are available for this example, data and sensitivity tests are available in Le Boursicaud et al. (2016). Given that image velocimetry techniques perhaps offer the best opportunity to estimate flows under these extreme conditions, researchers interested in reconstructing flash flood processes may find it valuable to understand how the range of available methods perform relative to each other, and software developers may find it instructive to consider how newly developed techniques compare with existing approaches under a diverse range of flow conditions.

**RC 2.4:** Section 2.9 River Tiber, Italy (Page 10, Ln 13-24): In my opinion, the single measurement of average velocity, which is provided as validation data for this site has limited value for the comprehensive analysis of different image velocimetry techniques reliability and efficiency. Please, provide explicit reasoning why this data will also help to meet the declared dataset objectives.
**AC 2.4:** Whilst only a single reference velocity value is available for the Tiber case-study, this measurement is representative of the surface velocities within an area of approx. 3 x 3m. The RVM20 speed surface radar system measurements can be compared with outputs derived from image velocimetry analysis within the 3 x 3m footprint. Similar to the St-Julien torrent case, this data set also represents a moderate flood event captured in February 2015. Images of floods suitable for velocimetry analysis are typically very rare and we believe that researchers and software developers may find this data set valuable to refine their algorithms and procedures.

**RC 2.5:** Dataset: I have realized that for some sites (e.g., Arrow River, Bradano River), scenes are not aligned with each other, i.e., ground (river banks) is not stable. In my opinion, key point alignment is needed to simplify the use of the dataset. This way, if the ground is stable for all the scenes, optical flow techniques can be easily implemented out-of-the-box for velocity field estimation.
**AC 2.5:** We acknowledge that the image sequences for the Arrow River and Bradano River case-studies are not stabilised. This is one of the critical challenges of using mobile platforms for image velocimetry analysis and the preferred approach may vary from researcher-to-researcher. Differences in the stabilisation technique may also have implications on the subsequent velocity outputs. Therefore, we deliberately chose to omit this stage and leave this to the discretion of the author. However, in the revised manuscript we will provide both the raw footage and a stabilised version using our preferred stabilisation method. The manuscript text will also be updated to reflect the addition of these stabilised frames.

**RC 2.6:** Dataset: I recommend authors to consider the change of format for the provided images to GeoTIFF (or similar) to provide explicit georeferencing capabilities. It will substantially simplify the validation procedure by providing a solid basis for validation data georeferencing.

**AC 2.6:** Unfortunately, the geographical coordinates are unknown for many of the image sequences presented. In the majority of cases, the ground control points were surveyed using an instrument that utilises a local reference (e.g. total station).

**RC 2.7:** Dataset: I did not find any validation data mentioned in the manuscript (Section 2) in the provided dataset archive.

**AC 2.7:** In response to comments from Reviewers 1, 2 and 3, the revised version of the manuscript will fully describe the validation data available for each of the case studies. This validation data will be collated, standardised and published in the data archive along with the orthorectified images, which were presented in the original submission.

**RC 2.8:** Dataset: In my opinion, the additional section, which will confirm the introduced dataset validity and its corresponding value for the target community, is needed. The potential reader has to be sure that the dataset is consistent with the declared objectives and therefore serves the reader's needs the best (e.g., benchmarking the new technique/software). I recommend authors to provide a brief analysis of the single case study showing the extracted velocities and comparing them to the validation data. Authors also may consider supporting the corresponding analysis with a code example - this may significantly increase the reader's interest to the dataset and manuscript itself.

**AC 2.8:** Analysis of the datasets provided is beyond the scope of this Data Description paper but we invite the reviewer to explore the references cited within the sub-section of each case-study and Table A2 as the dataset presented within this manuscript have been utilised to generate flow velocity data in previous work.

**Reviewer 3 Comments (RC3)**

**RC 3.1:** I think that the work is valuable and interested in the hydrology community for the development of image-based techniques, which could be further applied in modeling and monitoring.

**AC 3.1:** We would like to thank the reviewer for taking the time to assess the suitability of this manuscript to be published in Earth System Science Data, and for the constructive comments provided.

**RC 3.2:** However, it is not clear to me what contributions this paper offers. The abstract mentions inter-comparison and validation of the various techniques, but they were not actually performed, which seems to be missing a major component of the paper.

**AC 3.2:** The purpose of this manuscript is to introduce datasets that can be used for inter-comparison and validation of various techniques, rather than to perform inter-comparisons. This is beyond the scope of a Data Description paper. Currently, there exist several non-intrusive flow measurement techniques, and new ones being further developed. Performance tests of such techniques require the availability of optical flow data with reference measurements. Collection of such data is a laborious process and requires special, often expensive, equipment. This equipment is not necessarily available to every researcher who develops algorithms of flow analysis through image processing. Our goal is to facilitate further development and comparative tests of new and existing non-intrusive flow measurement techniques by making the necessary test data readily available to every researcher. We invite the reviewer to explore the references cited within the sub-section of each case-study and Table A2 as the datasets presented within this manuscript have been utilised to generate flow velocity data in previous work.

**RC 3.3:** Abstract (Page 1, Ln 10): It is mentioned that 13 case studies have been presented in the dataset, but Section 2 describes only 12.

**AC 3.3:** This is a typographical error and will be corrected. The abstract should read: 'Validation data is available for 11 of the 12 case studies presented enabling these data to be used for validation and accuracy assessment'.

**RC 3.4:** The validation data exists for most cases, then why not present the resulting datasets in the form that is directly compared and validate, instead of the image clips?

**AC 3.4:** The revised version of the manuscript will fully describe the validation data available for each of the case studies. This validation data will be collated and published in the data archive along with the orthorectified images, which were presented in the original submission.

**RC 3.5:** Even if quantitative validation is addressed, the measurements are taken at specific time and location of the river (i.e. specific hydro-geomorphic setting), so it may not be comparable if someone uses different camera and processing technique at different time and/or location. I understand that the nature of the observation and approach is not suitable for generalization, but the paper in the current form doesn't seem to fit into the context of "towards harmonization of the techniques"

**AC 3.5:** The purpose of our approach is indeed specific to a particular instance and location within the river. By ensuring that images are acquired at the same time (or river stage) as the reference measurements, a comparison between the two approaches will be possible. Furthermore, this database seeks to present examples from a range of hydro-geomorphic settings, which will enable researchers to assess the suitability of their chosen approach under hydrological conditions that are of particular interest to them.

---

## Author Response (AR1)

**Reviewer 1 Comments**

**Reviewers Comment (RC) 1:** This contribution is unique and important for the society of research and development on the image-based hydrometry approach. Provided images are useful for the RD mentioned above. Other than images, information for validation is described in the manuscript. The unfortunate thing is that quantitative validation data was not included in the provided data set. This restricts the aim of this contribution, "validation and accuracy assessment" (the last sentence in an abstract.) To accomplish the objective of the study further, I suggest some modifications in both the manuscript and the dataset.

**Authors Comment (AC) 1:** Thanks for the considered review of our submitted article. In response to comments from reviewers 1, 2 and 3, the revised version of the manuscript now fully describes the validation data available for each of the case studies. This validation data has been collated and published in the data archive along with the orthorectified images.

**RC 2:** Lines 11-14, page 2: Two sentences are discussing the image velocimetry application in labs and the logical flow between two sentences is difficult to follow. To make this part easier for reading, one option is to move "wide variety of experimental conditions" at the beginning part of the second sentence, since this part is a distinguishing point to the first sentence

**AC 2:** The sentences in question has been revised to read: 'Image velocimetry involves the application of cross-correlation, or computer vision techniques on a series of consecutive images (or extracted video frames) to generate vectors of water velocities across a field-of-view. It was originally developed for use in highly controlled laboratory settings. However, since its original conception, its application has expanded from use in the laboratory (e.g., Dudderar and Simpkins, 1977; Adrian, 1984; Pickering and Halliwell, 1984), to include a wide variety of experimental conditions.'

**RC 3:** Figure 1. I suggest dropping "Geographical" from the caption or add some more information regarding geography in the figure, e.g. water network, river basin, elevation, etc.

**AC 3:** The Figure caption has been modified to read: 'Locations of the monitoring sites from which data is presented. Numbers correspond with the in-text subsections: (1) River Arrow, UK; (2) River Thalhofen, Germany; (3) Murg River, Switzerland; (4) Alpine River, Austria; (5) River Brenta, Italy; (6) La Morge, France; (7) St-Julien torrent, France; (8) River La Vence, France; (9) River Tiber, Italy; (10) River Bradano, Italy; (11) River Noce, Italy. Not shown: (12) Castor River, Canada; (13) Salmon River, Canada. Map spatial reference: ETRS (1989).'

**RC 4:** Figures 2 and 3. Original and rectified images are provided in each figure and I guess the directions are rotated. Better to indicate the direction of the flow, e.g. by putting the arrow with a label of "flow" onto each panel.

**AC 4:** For both Figures 2, 3, and 4, the flow direction has been indicated using an arrow along with the word 'Flow'.

**RC 5:** Table A2. Label, this is quite a minor thing but I suggest use "Image Aquisition" instead of "Data Aquisition" in the label.

**AC 5:** The label of Table A2 has been modified to read: 'Image Acquisition'.

**RC 6:** Table A2. Validation data, I suggest to add the description about validation data (e.g. how and where).

**AC 6:** In order to support the presentation of the validation data we have used this part of the table to provide a summary of the reference measurements undertaken for each case-study (e.g. instrument, number of points, location, etc.).

**RC 7:** Table A2. Flow information, I suggest adding the mean velocity, representative depth, Froude number, width etc. (maybe, rotate 90 degrees the table to expand the width of the table).
**AC 7:** Unfortunately, this information is not readily available for all the case-studies. However, we will have ensured that all relevant information describing the hydrological conditions for each case-study is presented within each sub-section of the text.

**RC 8:** Data-set. Better to include movie file for each site for making easier to know the image characteristics and image recording approaches. (I made by myself for the purpose of review, and I can share it if needed.) Also suggested is providing text file(s) specifying the image resolution, location of the edges of images, and frame rate, and/or provide e.g. jgw, tfw and pgw files for corresponding image/folder (for jpg, tiff and png image, respectively).
**AC 8:** Videos consisting of the orthorectified imagery has been provided in the Data Archive. These videos have been produced at the same image resolution as the orthorectified/stabilised images. This information is provided in the Readme.txt associated with each case study within the Data submission. Unfortunately, the geographical coordinates are unknown for many of the image sequences presented. In most cases, the ground control points were surveyed using an instrument that utilises a local reference (e.g. total station). In these cases, it would be inappropriate to provide a tgw, tfw, etc. file.

**RC 9:** Data-set. For sites with velocity distribution measured for validation, provide the location and velocity of the data as e.g. CSV file.
**AC 9:** The revised submission now includes reference velocity measurements for each of the case studies where it is available. This validation data has been collated and published in the data archive along with the orthorectified images. The locations of the velocity measurements are provided in pixel units based on the orthorectified images.

**RC 10:** Data-set. Type of image file and the structure of file name differ for each folder, this making the pre-processing a bit troublesome for a potential user of the data. Could you provide also a unified formatted image set, e.g. 0000.png? (I made this also by myself for review, and I can share it if needed.)
**AC 10:** The filenames have been altered and are now presented in a standardised format e.g. 00001.png.

**Reviewer 2 Comments**

**RC 1:** The present manuscript is aimed to introduce the new dataset, which will help to systematize and benchmark the emerging techniques for image-based river surface velocity estimation. The corresponding dataset consists of pre-processed videos from 12 research sites located in six different countries and covered a wide range of fluvial settings. In my opinion, the introduced dataset has sound potential and of high interest in the research community. However, I recommend authors to provide major revisions which may help to increase the dataset value for the target community and make it the first benchmark dataset for image-based velocimetry techniques (e.g., as the MNIST database for image classification).
**AC 1:** We would like to thank the reviewer for taking the time to provide a thorough review of our submitted manuscript.

**RC 2:** Abstract (Page 1, Ln 10): It is mentioned that 13 case studies have been presented in the dataset, but Section 2 describes only 12.

**AC 2:** 13 case studies are now presented and the abstract now reads: 'Reference data is available for 12 of the 13 case studies presented enabling these data to be used for reference and accuracy assessment.'

**RC 3:** Section 2.7 St-Julien torrent, France (Page 8, Ln 24-31): As for this particular case study, the validation data is unavailable, the explicit description is needed to clarify the reasons behind the inclusion of the corresponding data to the introduced dataset. At least, it is not clear how this data will help to pursue one of the dataset objectives as "testing specific image velocimetry techniques."

**AC 3:** This particular case study represents a flash flood, which occurred in a torrent system in Italy producing mean velocities of approx. 6 m s$^{-1}$. Whilst no reference measurements are available for this example, image velocimetry techniques perhaps offer the best opportunity to estimate flows under these extreme conditions. Researchers interested in reconstructing flash flood processes may find it valuable to understand how the range of available methods perform relative to each other, and software developers may find it instructive to consider how newly developed techniques compare with existing approaches under a diverse range of flow conditions. This justification has been provided in the manuscript.

**RC 4:** Section 2.9 River Tiber, Italy (Page 10, Ln 13-24): In my opinion, the single measurement of average velocity, which is provided as validation data for this site has limited value for the comprehensive analysis of different image velocimetry techniques reliability and efficiency. Please, provide explicit reasoning why this data will also help to meet the declared dataset objectives.

**AC 4:** Whilst only a single reference velocity value is available for the Tiber case-study, this measurement is representative of the surface velocities within an area of approx. 3 x 3m. The RVM20 speed surface radar system measurements can be compared with outputs derived from image velocimetry analysis within the 3 x 3m footprint. This justification has been provided in the manuscript.

**RC 5:** Dataset: I have realized that for some sites (e.g., Arrow River, Bradano River), scenes are not aligned with each other, i.e., ground (riverbanks) is not stable. In my opinion, key point alignment is needed to simplify the use of the dataset. This way, if the ground is stable for all the scenes, optical flow techniques can be easily implemented out-of-the-box for velocity field estimation.

**AC 5:** We acknowledge that the original submission of image sequences for the Arrow River and Bradano River case-studies were not stabilised. In the revised submission we have included both the stabilised and original images. We have chosen to include the un-stabilised footage as stabilisation is one of the critical challenges of using mobile platforms for image velocimetry analysis and the preferred approach may vary from researcher-to-researcher. Differences in the stabilisation technique may also have implications on the subsequent velocity outputs.

**RC 6:** Dataset: I recommend authors to consider the change of format for the provided images to GeoTIFF (or similar) to provide explicit georeferencing capabilities. It will substantially simplify the validation procedure by providing a solid basis for validation data georeferencing.

**AC 6:** Unfortunately, the geographical coordinates are unknown for many of the image sequences presented. In most cases, the ground control points were surveyed using an instrument that utilises a local reference (e.g. total station). Therefore, georeferencing of the images is problematic.

However, the reference velocity data has been provided in pixel units to ensure that comparisons between image velocimetry outputs and reference measurements should be straightforward.

**RC 7:** Dataset: I did not find any validation data mentioned in the manuscript (Section 2) in the provided dataset archive.

**AC 7:** The revised version of the manuscript now fully describes the validation data available for each of the case studies. This validation data has been collated and published in the data archive along with the orthorectified images, which were presented in the original submission.

**RC 8:** Dataset: In my opinion, the additional section, which will confirm the introduced dataset validity and its corresponding value for the target community, is needed. The potential reader has to be sure that the dataset is consistent with the declared objectives and therefore serves the reader's needs the best (e.g., benchmarking the new technique/software). I recommend authors to provide a brief analysis of the single case study showing the extracted velocities and comparing them to the validation data. Authors also may consider supporting the corresponding analysis with a code example - this may significantly increase the reader's interest to the dataset and manuscript itself.

**AC 8:** Analysis of the datasets provided is beyond this scope of this Data Description paper but we invite the reviewer to explore the references cited within the sub-section of each case-study and Table A2 as the datasets presented within this manuscript have been utilised to generate flow velocity data in previous published work.

**Reviewer 3 Comments**

**RC 1:** I think that the work is valuable and interested in the hydrology community for the development of image-based techniques, which could be further applied in modeling and monitoring.

**AC 1:** We would like to thank the reviewer for taking the time to assess the suitability of this manuscript to be published in Earth System Science Data, and for the constructive comments provided.

**RC 2:** However, it is not clear to me what contributions this paper offers. The abstract mentions inter-comparison and validation of the various techniques, but they were not actually performed, which seems to be missing a major component of the paper.

**AC 2:** The purpose of this manuscript is to introduce datasets that can be used for inter-comparison and validation of various techniques, rather than to perform inter-comparisons. This is beyond the scope of a Data Description paper. However, we invite the reviewer to explore the references cited within the sub-section of each case-study and Table A2 as the datasets presented within this manuscript have been utilised to generate flow velocity data in previous work.

**RC 3:** Abstract (Page 1, Ln 10): It is mentioned that 13 case studies have been presented in the dataset, but Section 2 describes only 12.

**AC 3:** This was a typographical error. However, we have since added an additional case-study. Therefore, we keep the text as it was in the original manuscript.

**RC 4:** The validation data exists for most cases, then why not present the resulting datasets in the form that is directly compared and validate, instead of the image clips?

**AC 4:** The revised version of the manuscript now fully describes the validation data available for each of the case studies. This validation data has been collated and published in the data archive along with the orthorectified images.

**RC 5:** Even if quantitative validation is addressed, the measurements are taken at specific time and location of the river (i.e. specific hydro-geomorphic setting), so it may not be comparable if someone uses different camera and processing technique at different time and/or location. I understand that the nature of the observation and approach is not suitable for generalization, but the paper in the current form doesn't seem to fit into the context of "towards harmonization of the techniques"

**AC 5:** The purpose of our approach is indeed specific to a particular instance and location within the river. By ensuring that images are acquired at the same time (or river stage) as the reference measurements, a comparison between the two approaches will be possible. Furthermore, this database seeks to present examples from a range of hydro-geomorphic settings, which will enable researchers to assess the suitability of their chosen approach under hydrological conditions that are of particular interest to them.

[revised manuscript text omitted]

---

## Author Response (AR2)

**Reviewer 1 Comments**

**Reviewers Comment (RC) 1:** The work presented here is unique and potentially useful for the development of the image velocimetry techniques. The comments from the reviewers seem to be well addressed. The quality and quantity of the provided data are upgraded. In conclusion, I'm supportive of acceptance as is.

Authors Comment (AC) 1: Thank you for your considered reviews, which greatly improved the quality of this submission.

**Reviewer 2 Comments**

**RC 1:** I acknowledge that my comments on the original manuscript submission have been fully considered in the revised version of the manuscript. In my opinion, the revised manuscript can be accepted subject to technical and editorial corrections.

**AC 1:** Thank you for your considered reviews, which greatly improved the quality of this submission.

**Reviewer 3 Comments**

**RC 1:** This paper compiles imagery for image velocimetry analysis and reference data from 13 observations over the Europe and Canada. I understand that this is a data description paper, and the main objective is to document the current situation where a range of equipment and analysis method as well as environmental setting exists. However, it is not clear what the limitations or efficacy of different approaches are, despite the details provided for the individual measurements. Does the resolution of image matter? Pre-processing software seems to be tailored to the choice of equipment or camera angle, then does the environment setting or nature of flow determine those choice? How would you suggest inter-comparison using this dataset as a benchmark? Use the same camera but different software or use different equipment for the same site—is that applicable to the setting? I think that the objective of this paper is relevant to the hydrological community, but the paper in its current form does not offer insights bridging into the next step (i.e. organized inter-comparison).

**AC 1:** We would like to thank the reviewer for taking the time to assess the suitability of this manuscript to be published in Earth System Science Data, and for the constructive comments provided. We have addressed these comments through the addition of text within the Introduction and Conclusion sections of the manuscript, as highlighted in the marked-up version within this pdf document.

**Towards harmonization of image velocimetry techniques for river surface velocity observations**

Matthew. T. Perks1, Silvano Fortunato Dal Sasso2, Alexandre Hauet3, Elizabeth Jamieson4, Jérôme Le Coz5, Sophie Pearce6, Salvador Peña-Haro7, Alonso Pizarro2, Dariia Strelnikova8, Flavia Tauro9, James Bomhof4, Salvatore Grimaldi9,10, Alain Goulet4, Borbála Hortobágyi1, Magali Jodeau11,12, Sabine Käfer13, Robert Ljubičić14, Ian Maddock6, Peter Mayr15, Gernot Paulus8, Lionel Pénard5, Leigh Sinclair4, and Salvatore Manfreda16

1School of Geography, Politics and Sociology, Newcastle University, Newcastle upon Tyne, United Kingdom, 2Department of European and Mediterranean Cultures: Architecture, Environment and Cultural Heritage (DiCEM), University of Basilicata, 75100 Matera, Italy. 3Electricité de France, DTG, Grenoble, France. 4National Hydrological Services, Environment and Climate Change Canada. 5INRAE, UR RiverLy, River Hydraulics, Villeurbanne, France. 6School of Science and the Environment, University of Worcester, Worcester, UK. 7Photrack AG: Flow Measurements, Ankerstrasse 16a, 8004 Zürich, Switzerland. 8School of Geoinformation, Carinthia University of Applied Sciences, 9524 Villach, Austria. 9Department for Innovation in Biological, Agro-food and Forest Systems, University of Tuscia, Viterbo, Italy. 10Department of Mechanical and Aerospace Engineering, Tandon School of Engineering, New York University, Brooklyn, NY, United States. 11Electricité de France, R&D, Chatou, France 12LHSV, Chatou, France. 13Verbund Hydro Power GmbH, 9500 Villach, Austria. 14Faculty 
[revised manuscript text omitted]